# Children exhibit superior memory for attended but outdated information compared to adults

Yingtao Fu[1,4], Tingyu Guo[1,4], Jiewei Zheng [1], Jie He [1] ✉, Mowei Shen [1] ✉ & Hui Chen [1,2,3] ✉

Research on the development of cognitive selectivity predominantly focuses on attentional selection. The present study explores another facet of cognitive selectivity—memory selection—by examining the ability to filter attended yet outdated information in young children and adults. Across five experiments involving 130 children and 130 adults, participants are instructed to use specific information to complete a task, and then unexpectedly asked to report this information in a surprise test. The results consistently demonstrate a developmental reversal-like phenomenon, with children outperforming adults in reporting this kind of attended yet outdated information. Furthermore, we provide evidence against the idea that the results are due to different processing strategies or attentional deployments between adults and children. These results suggest that the ability of memory selection is not fully developed in young children, resulting in their inefficient filtering of attended yet outdated information that is not required for memory retention.

Given the limited cognitive resources available for human online processing[1–3], information selection is crucial for navigating and surviving in this information-rich world. Selective attention has long been regarded as the primary mechanism for prioritizing relevant information while filtering out the irrelevant[4–7]. Developmentally, the ability of selective attention gradually matures from childhood to adulthood[8,9]. Infants and young children often struggle to concentrate on specific information and filter out distractions. However, individuals demonstrate significant improvements in both focusing and filtering as they mature. Consequently, while adults typically excel at processing task-relevant information, it is often observed that children outperform adults in processing task-irrelevant information[10–13].

It has been believed that attention not only determines what we perceive, it also determines what we remember. Traditionally, attention has been proposed as the gateway to working memory, as it determines what information is selected into working memory[14–16]. That is, the external information that is attended to would be automatically incorporated into the working memory system. However, recent attribute amnesia studies challenged this assumption by showing that even fully attended information could be blocked out of working memory[17–21]. For example, in one typical attribute amnesia experiment, participants were asked to find a target letter among distractor digits and report its location. In this task set, the identity of the target letter served as a key feature, which needed to be attended and used for detecting and localizing the target, but was not necessarily to be memorized for later report. Therefore, the key feature is a kind of attended yet outdated information. In such a case, when participants were unexpectedly probed about the key feature, they were unable to report it accurately. Subsequent studies found that the report failure of the key feature was due to the lack of memory consolidation of this attended information[22–24]. These studies reveal that the information selected by attention is not necessarily selected into working memory; instead, working memory has a reselection process for information that had been fully attended. This process is termed as memory selection to distinguish it from attentional selection[24,25]. The memory selection enables our brain to filter out the attended yet

[1]Department of Psychology and Behavioral Sciences, Zhejiang University, Hangzhou, China. [2]The First Affiliated Hospital, Zhejiang University School of Medicine, Hangzhou, China. [3]The State Key Lab of Brain-Machine Intelligence, Zhejiang University, Hangzhou, China. [4]These authors contributed equally: Yingtao Fu, Tingyu Guo. ✉e-mail: jiehe@zju.edu.cn; mwshen@zju.edu.cn; chenhui@zju.edu.cn

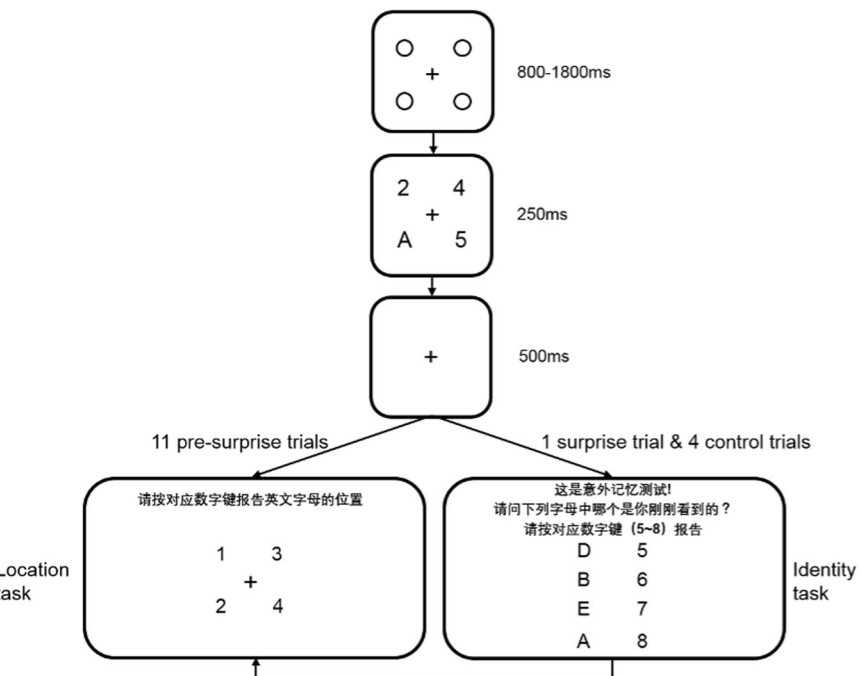

**Fig. 1 | Sample trial sequence in Experiment 1.** The questions were presented in Chinese and the translations were: (1) location question: Press a corresponding number to indicate the location of target letter; (2) identity surprise question: This is a surprise memory test! Which one of following four letters is the one that you had just seen? Press a corresponding number (5-8) to indicate its identity. The corresponding numbers (1-4) in the location task were pre-defined and replaced by four placeholders during the experiment. The adults read the question and responded by pressing corresponding keys, whereas children were asked by the experimenter and made verbal responses.

outdated information, constituting an efficient information-selection system together with attentional selection.

Despite extensive research on the development of attentional selection[8–13], the development of memory selection remains largely unexplored. In this study we directly investigated this issue by comparing the ability of memory selection between young children and adults, on the basis of attribute amnesia paradigm. Specifically, we compared the performance between adults and children in reporting the key feature in attribute amnesia tasks. Since both groups needed to attend to and use the key feature to complete such tasks, the performance difference in reporting the key feature between adults and children would reflect their different abilities of memory selection rather than attentional selection.

In Experiment 1, participants were asked to find a target letter among three distractor digits and report its location. In one critical trial (i.e., surprise trial), they were unexpectedly asked to report the identity of the target letter (i.e., the key feature). Surprisingly, children performed significantly better than adults in reporting the identity of the target letter. Experiment 2 generalized this finding by using stimuli that were highly familiar (i.e., animal pictures) to both adults and children, suggesting that the results reflected developmental disparities in memory selectivity, rather than variations in the familiarity of stimuli across different groups. Experiment 3 ruled out an alternative explanation that adults' worse performance in reporting the target identity might be due to the fact that they performed the task based on categorization without identifying the target. Experiment 4 adopted eye-tracking technology revealing that adults indeed allocated more attention to the target containing the key feature, yet their performance in reporting the key feature was still worse than children. This result ruled out the possibility that the previous findings were driven by the different allocation of attention during the task between two groups. Experiment 5 further confirmed the main findings with a larger sample size. These results consistently showed that compared with adults, children had a better memory for attended information

that was not necessarily to be maintained. This developmental reversal-like phenomenon suggests that the ability of memory selection is not fully developed for young children, which leads them to memorize more attended but outdated information than adults.

## Results
### Experiment 1
Experiment 1 adopted the attribute amnesia paradigm and compared the performance between adults and children in reporting the key feature in the surprise trial. If children have difficulty in memory selection of attended information, they would show a better memory for the key feature than adults.

As shown in Fig. 1, each trial began with a centered black fixation cross among four black placeholder circles. After that, the stimulus array appeared for 250 ms, containing one English letter target (A, B, D, or E) and three Arabic number distractors (2 to 9). In the first 11 pre-surprise trials, participants were asked to report the location of the target letter (location task). In the 12th trial (i.e., the surprise trial), prior to the location task, participants were unexpectedly presented with a forced-choice question requiring them to indicate which of four letters was the target letter they had just seen. Following the surprise trial, participants completed four control trials which were in the same format as the surprise trial. The adults responded by pressing corresponding keys on a keyboard, whereas children made verbal responses that were recorded by the experimenter. Twenty adults ($M_{age} = 18.95$ years) and twenty children ($M_{age} = 5.75$ years) participated in this experiment.

The results are shown in Table 1. The location report accuracy on the pre-surprise trials for adults and children were 99% and 94% respectively, indicating that both adults and children could accurately locate the target letter presented among the distractor numbers. For the question in the surprise trial, only 10 of 20 (50% correct) adults correctly reported the identity of the target letter, which was significantly worse than the accuracy in the first control trial (50% vs.

**Table 1 | Results of Experiment 1 ($N = 20$ for each age group)**

|              | Pre-surprise | Surprise | Control1 | Control2 | Control3 | Control4 |
|--------------|--------------|----------|----------|----------|----------|----------|
| *Children*   |              |          |          |          |          |          |
| Location     | 94%          | 60%      | 60%      | 60%      | 85%      | 95%      |
| Identity     | N/A          | 80%      | 65%      | 80%      | 80%      | 95%      |
| *Adults*     |              |          |          |          |          |          |
| Location     | 99%          | 90%      | 80%      | 90%      | 100%     | 95%      |
| Identity     | N/A          | 50%      | 90%      | 100%     | 90%      | 100%     |

Note. *N/A* Not applicable. Source data are provided as a Source Data file.

**Table 2 | Results of Experiment 2 ($N = 20$ for each age group)**

|              | Pre-surprise | Surprise | Control1 | Control2 | Control3 | Control4 |
|--------------|--------------|----------|----------|----------|----------|----------|
| *Children*   |              |          |          |          |          |          |
| Location     | 99%          | 65%      | 85%      | 90%      | 90%      | 95%      |
| Identity     | N/A          | 85%      | 90%      | 100%     | 100%     | 100%     |
| *Adults*     |              |          |          |          |          |          |
| Location     | 99%          | 95%      | 95%      | 90%      | 95%      | 95%      |
| Identity     | N/A          | 55%      | 95%      | 100%     | 100%     | 100%     |

Note. N/A Not applicable. Source data are provided as a Source Data file.

90%), $\chi^2(1, N = 40) = 7.619$, $p = 0.006$, $\varphi = 0.436$. As a contrast, 16 of 20 (80% correct) children correctly reported the identity of the target letter, which showed no statistically significant difference from that in the first control trial (80% vs. 65%), $\chi^2(1, N = 40) = 1.129$, $p = 0.288$, $\varphi = 0.168$, $BF_{10} = 0.572$; as well as from that in the following control trials (80% vs. 80%; 80%; 95%), $\chi^2(1, N = 40) \leq 2.057$, $ps \geq 0.151$, $\varphi \leq 0.227$, $BF_{10} \leq 0.631$. Importantly, children performed significantly better than adults in reporting the identity of the target letter (80% vs. 50%), $\chi^2(1, N = 40) = 3.956$, $p = 0.047$, $\varphi = 0.314$. Notably, as both adults and children had to attend to and use the target identity to locate the target in this task set, this finding indicated that children had not yet developed the ability of memory selection for the attended yet outdated information.

## Experiment 2

Experiment 1 revealed that children performed better than adults in reporting the target letter in the surprise test. Someone might argue that this result arose from the fact that children exerted more effort in processing the key feature (i.e., identity) due to their less familiarity with the stimuli, rather than reflecting developmental differences in memory selection. Therefore, Experiment 2 was designed to replicate and generalize the findings of Experiment 1 by adopting stimuli that were highly familiar (i.e., animal pictures) to both adults and children. We asked 20 additional children from the same kindergarten to select four animal pictures that they were most familiar with, out of eight common animals (chicken, dog, frog, rabbit, cat, pig, turtle, and goat). The four most familiar animals (chicken, dog, rabbit, and cat) were then used as targets in the attribute amnesia task for both children and adults. In this task, participants were required to locate a target animal among three clothing distractors (coat, hat, pants, shoe, sweater, or boot) in a 1000-ms stimuli presentation. Then, in the surprise test, participants were asked to report which animal they had just found. Twenty new adults ($M_{age} = 20.50$ years) and 20 new children ($M_{age} = 6.10$ years) participated in this experiment.

The results were consistent with Experiment 1. As shown in Table 2, only 11 of 20 (55% correct) adults were correct on the identity report task in the surprise trial, which significantly improved in the first control trial (55% vs. 95%), $\chi^2(1, N = 40) = 8.533$, $p = 0.003$, $\varphi = 0.462$. In contrast, 17 of 20 (85% correct) children correctly reported the identity of the target in the surprise trial, which showed no statistically

significant difference from that in the first control trial (85% vs. 90%), $\chi^2(1, N = 40) = 0.229$, $p = 0.633$, $\varphi = 0.076$, $BF_{10} = 0.282$; as well as from that in the following control trials (85% vs. 100%; 100%; 100%), $\chi^2(1, N = 40) = 3.243$, $ps = 0.072$, $\varphi = 0.285$, $BF_{10} = 0.806$. Importantly, the comparison between adults and children showed that children performed significantly better than adults in reporting the target identity in the surprise trial (55% vs. 85%), $\chi^2(1, N = 40) = 4.286$, $p = 0.038$, $\varphi = 0.327$. These results suggested that the memory selection for children was weaker than adults even for highly familiar stimuli.

## Experiment 3

Experiment 3 aimed to rule out two other possible factors that might contribute to children's better memory performance in the surprise test than adults. Firstly, it could be that adults located the target letter by category without resolving its identity, whereas children did not have this ability and had to access the identity of the target letter. To increase our confidence that both adults and children located the target by attending to and use the target identity, in Experiment 3 we asked participants to locate a target number larger than five among three distractor numbers smaller than five. Previous studies have revealed that this number comparison task demands direct access to numerical quantity for both adults and young children[26,27]. In Supplementary Methods, we provided direct evidence that participants did access the identity of the target number in the specific task set in our study. Secondly, it remained possible that adults' memory was impaired when reading the surprise question whereas this was not the case for children who were directly asked by the experimenter. In Experiment 3 both adults and children were directly asked by the experimenter in the surprise test.

This experiment was identical to Experiment 1 except as follows. Participants were asked to locate a target number that was larger than five (6, 7, 8, or 9) among three distractor numbers smaller than five (1, 2, 3, or 4). As this task was more difficult than Experiment 1, the stimulus presentation time was prolonged to 1000 ms. In the surprise test, participants were asked to report the identity of the target number. In addition, unlike in Experiment 1 wherein adults read the surprise question themselves, here the experimenter directly asked adults to indicate the target number in the surprise test. Twenty new adults ($M_{age}=24.20$ years) and 20 new children ($M_{age}=5.95$ years) participated in this experiment.

**Table 3 | Results of Experiment 3 (*N* = 20 for each age group)**

|  | Pre-surprise | Surprise | Control1 | Control2 | Control3 | Control4 |
|---|---|---|---|---|---|---|
| *Children* |  |  |  |  |  |  |
| Location | 94% | 65% | 65% | 70% | 80% | 70% |
| Identity | N/A | 90% | 90% | 90% | 85% | 70% |
| *Adults* |  |  |  |  |  |  |
| Location | 97% | 80% | 85% | 95% | 95% | 95% |
| Identity | N/A | 60% | 90% | 100% | 100% | 100% |

Note. N/A= Not applicable. Source data are provided as a Source Data file.

**Table 4 | Results of Experiment 4 (*N* = 20 for each age group)**

|  | Pre-surprise | Surprise | Control1 | Control2 | Control3 | Control4 |
|---|---|---|---|---|---|---|
| *Children* |  |  |  |  |  |  |
| Location | 96% | 95% | 90% | 80% | 85% | 90% |
| Identity | N/A | 100% | 90% | 85% | 100% | 100% |
| *Adults* |  |  |  |  |  |  |
| Location | 98% | 80% | 95% | 95% | 100% | 100% |
| Identity | N/A | 50% | 100% | 100% | 100% | 100% |

Note. N/A= Not applicable. Source data are provided as a Source Data file.

The results were consistent with previous experiments. As shown in Table 3, only 12 of 20 (60% correct) adults were correct on the identity report task in the surprise trial, which significantly improved in the first control trial (60% vs. 90%), $\chi^2(1, N = 40) = 4.800$, $p = 0.028$, $\varphi = 0.346$. In contrast, 18 of 20 (90% correct) children correctly reported the identity of the target in the surprise trial, which showed no statistically significant difference from that in the first control trial (90% vs. 90%), $\chi^2(1, N = 40) = 0$, $p = 1.000$, $\varphi = 0$, $BF_{10} = 0.235$; as well as from that in the following control trials (90% vs. 90%; 85%; 70%), $\chi^2(1, N = 40) \leq 2.500$, $ps \geq 0.114$, $\varphi \leq 0.250$, $BF_{10} \leq 0.971$. Furthermore, the comparison between adults and children showed that children performed significantly better than adults in reporting the target identity in the surprise trial (60% vs. 90%), $\chi^2(1, N = 40) = 4.800$, $p = 0.028$, $\varphi = 0.346$. Therefore, the results, together with those in Experiments 1 and 2, provided converging evidence that children had a weaker ability of memory selection than adults, ruling out the aforementioned alternative explanations.

## Experiment 4
The results of Experiments 1-3 showed that children performed better in reporting the key feature than adults, suggesting that children had an underdeveloped memory selection. However, someone might argue that the better performance of children could be due to that children directed more attention on the target, and thus had a better memory for all the features (including the key feature) pertaining to the target. We believe that this alternative explanation could less likely be true, because previous studies found that compared with adults, children tended to allocate more attention on distractors rather than on the target[12,28–31]. Nonetheless, as there remained methodological differences between the current study and previous studies, to further exclude this possibility, we monitored participants' eye movements in Experiment 4 and adopted a similar task as Experiment 3. Twenty new adults (M_{age} = 19.80 years) and 20 new children (M_{age} = 6.00 years) participated in this experiment.

The behavior results were consistent with Experiment 3. As shown in Table 4, only 10 of 20 (50% correct) adults were correct on the identity report task in the surprise trial, which significantly improved in the first control trial (50% vs. 100%), $\chi^2(1, N = 40) = 13.333$, $p < 0.001$, $\varphi = 0.577$. In contrast, 20 of 20 (100% correct) children correctly reported the identity of the target in the surprise trial, which showed

no statistically significant difference from that in the first control trial (100% vs. 90%), $\chi^2(1, N = 40) = 2.105$, $p = 0.147$, $\varphi = 0.229$, $BF_{10} = 0.382$; as well as from that in the following control trials (100% vs. 85%; 100%; 100%), $\chi^2(1, N = 40) \leq 3.243$, $ps \geq 0.072$, $\varphi \leq 0.285$, $BF_{10} \leq 0.806$. Furthermore, the comparison between adults and children showed that children performed significantly better than adults in reporting the target identity in the surprise trial (50% vs. 100%), $\chi^2(1, N = 40) = 13.333$, $p < 0.001$, $\varphi = 0.577$.

The eye movement data analysis was focused on the fixation distribution during the stimulus array in pre-surprise trials. As shown in Fig. 2, for adults, the proportion of fixation duration on the target area was about six times as on each distractor area (57% vs. 9%), $t(19) = 13.652$, $p < 0.001$, Cohen's $d = 3.053$, 95% CI for Mean Difference = [40%, 55%]. As a contrast, for children, the proportion of fixation duration on the target area was only about two times as on each distractor area (40% vs. 19%), $t(19) = 11.123$, $p < 0.001$, Cohen's $d = 2.487$, 95% CI for Mean Difference = [18%, 26%]. Importantly, the between-group comparison showed that the proportional difference of fixation duration between target area and distractor area significantly differed between adults and children, $F(1, 38) = 42.293$, $p < 0.001$, $\eta_p^2 = 0.527$. This analysis unveiled that children tended to fixate more on distractors compared to adults, while adults exhibited a greater fixation on the target relative to children. This observation was consistent with previous research highlighting the tendency for adults to adopt a focused attentional mode, contrasting with the tendency for children to exhibit a broader, less focused attentional mode[10–13,30]. Although eye-tracking technology could not directly track the attentional deployment at the feature level, it was less likely that the children paid more attention on the target identity (key feature) in this case, given the fact that they showed less (rather than more) attentional deployment on the target stimulus. Thus, the current experiment provided evidence that the different performance of reporting key feature reflected the different ability of memory selection, rather than different attentional deployment between the two groups.

## Experiment 5
The sample size in Experiments 1-4 was relatively small, with 20 participants in each age group. Despite observing consistent results across four experiments, we recognized potential power issues in small

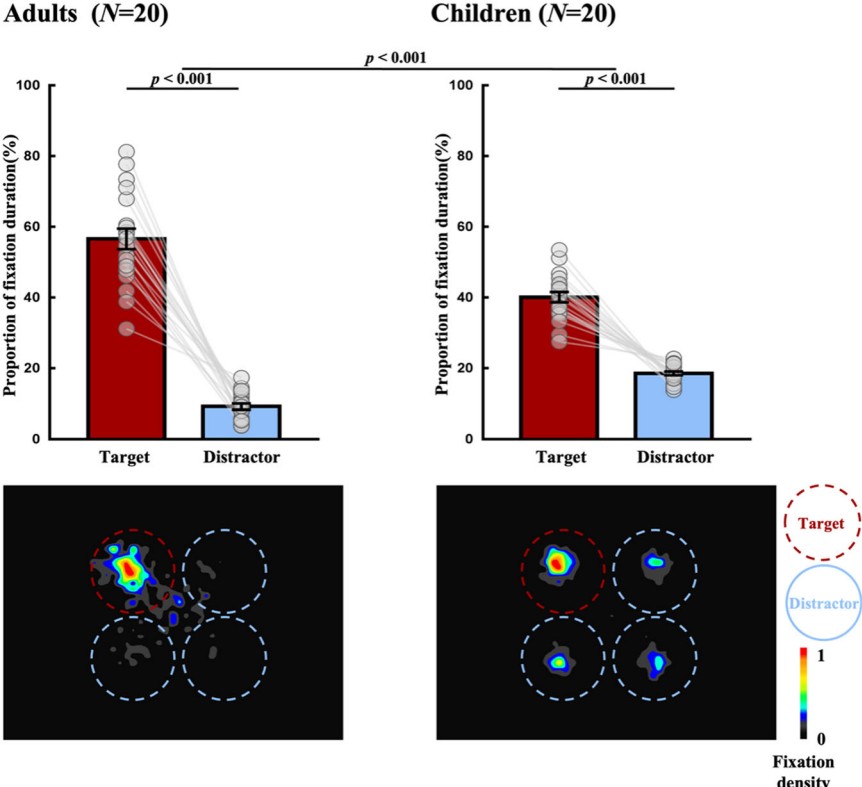

**Fig. 2 | Eye movement results of Experiment 4 for adults (N = 20) in left panel and children (N = 20) in right panel.** In histograms, data for the proportion of fixation duration on distractor were the average of the three distractors. Statistical tests were conducted using two-sided $t$ and $F$ tests. The heat maps demonstrated the density of fixation duration during the stimulus array, where the data from each trial were rotated as if the target location was on the top left. All heat maps were smoothed by a Gaussian filter[58]; the kernel size was set at 1, and the standard deviation (sigma) was set at 3. Error bars represent standard error of the mean. Source data are provided as a Source Data file.

sample studies (e.g., effect inflation, low reproducibility)[32]. To further ensure the robustness of our findings, we conducted a replication with a larger sample size of 50 participants in each group. This increased sample size was determined through a more conservative estimation of the effect size ($\varphi = 0.30$) regarding the performance difference between children and adults in reporting the key feature during the surprise test. This experiment was the same as Experiment 2. Fifty new adults ($M_{age} = 21.76$ years) and 50 new children ($M_{age} = 5.48$ years) participated in this experiment.

The results were consistent with previous findings. As shown in Table 5, only 23 of 50 (46% correct) adults were correct on the identity report task in the surprise trial, which significantly improved in the first control trial (46% vs. 98%), $\chi^2(1, N = 100) = 33.532$, $p < 0.001$, $\varphi = 0.579$. In contrast, 43 of 50 (86% correct) children correctly reported the identity of the target in the surprise trial, which showed no statistically significant difference from that in the first control trial (86% vs. 94%), $\chi^2(1, N = 100) = 1.778$, $p = 0.182$, $\varphi = 0.133$, $BF_{10} = 0.343$; as well as from that in the following control trials (86% vs. 94%; 88%; 94%), $\chi^2(1, N = 100) \leq 1.778$, $ps \geq 0.182$, $\varphi \leq 0.133$, $BF_{10} \leq 0.343$. Importantly, as previously found, children performed significantly better than adults in reporting the target identity in the surprise trial (86% vs. 46%), $\chi^2(1, N = 100) = 17.825$, $p < 0.001$, $\varphi = 0.422$. Furthermore, with the advantage of the large sample size, we were able to conduct an interaction analysis between groups, a procedure that typically demands a relatively large sample size in each group[33]. A Breslow-Day test of homogeneity showed that performance improvement from the surprise trial to the first control trial significantly differed between adults and children, $\chi^2(1) = 7.532$, $p = 0.006$, indicating a different pattern of attribute amnesia effect between the two groups. These results replicated our previous findings.

Another interesting finding was that children showed much worse performance on location reports in the surprise trial than adults (58% vs. 94%), which pattern had also been observed in Experiments 1-3. This observation suggested that the surprise test of identity elicited more disruption of location memory in children, implying that children's working memory storage might be more likely to be interfered with by unexpected events.

## Discussion

The current study consistently demonstrated a developmental reversal-like phenomenon: the children outperformed adults in reporting the key feature in an attribute amnesia task. These results suggest that the memory selection of attended information is not fully developed for young children, which leads them to be inclined to automatically encode attended but outdated information into memory.

The majority of prior research concerning the development of cognitive selectivity has concentrated on attentional selection[34]. Typically, young children have been observed to show a greater propensity for detecting or memorizing task-irrelevant information compared to adults. This tendency is often attributed to children's broader distribution of attention relative to adults[10-13,35,36]. For instance, in the classical flanker task, children's response to the target was significantly more influenced by distractors compared to adults[35]. Similarly, in a study employing change-detection and visual search tasks, children exhibited better recognition of irrelevant information than adults in both tasks[12]. Moreover, in a working memory study where participants were instructed to remember the color rather than the shape of various objects, children tended to encode irrelevant shapes to a greater extent than adults[36]. Consistent with these findings, the eye movement results from Experiment 4 also demonstrated that

**Table 5 | Results of Experiment 5 (N = 50 for each age group)**

|  | Pre-surprise | Surprise | Control1 | Control2 | Control3 | Control4 |
|---|---|---|---|---|---|---|
| *Children* | | | | | | |
| Location | 99% | 58% | 76% | 74% | 84% | 86% |
| Identity | N/A | 86% | 94% | 94% | 88% | 94% |
| *Adults* | | | | | | |
| Location | 99% | 94% | 98% | 100% | 98% | 100% |
| Identity | N/A | 46% | 98% | 100% | 100% | 100% |

Note. *N/A* Not applicable. Source data are provided as a Source Data file.

children allocated a higher proportion of attention to the distractors compared to adults.

The current study investigated the development of another aspect of cognitive selectivity—the ability of memory selection for attended information, and also found a weaker memory selection for children. The underdevelopment of both attentional selection and memory selection reflects a generally deficient selectivity in children's cognitive processing. Notably, the current finding implies that it should be cautious to investigate the ability of attentional selection through memory test paradigms. For example, children's better memory performance for task-irrelevant information has usually been thought to be due to their weaker attentional selection than adults; however, it remains possible that children and adults exist similar attentional processing on irrelevant information whereas children cannot block this information out of memory as efficiently as adults (i.e., weaker memory selection). One recent study provided a nice example of examining the development of attentional selectivity in a more direct manner, where participants' attention and their behavioral patterns were directly linked[37]. This is achieved by measuring attentional allocation via eye tracking during category learning, and then using these data to predict later categorization responses.

One recent study found that the attended yet outdated information (e.g., key feature) was blocked from working memory via a mechanism that actively inhibited this information from being stored, which was subject to executive control[23]. This is consistent with the theoretical framework proposing that inhibitory processes allow only information to enter working memory that is "along the goal-path", while suppress the activation of "off the goal-path" information and preventing such information from entering into working memory[38,39]. It has been generally found that children's inhibitory abilities are not yet mature enough to inhibit extraneous information compared with adults[40–46]. Therefore, we suspect that children's underdevelopment of memory selection might be due to the fact that they fail to inhibit the attended yet outdated information efficiently. This assumption is consistent with the development studies of cognitive inhibition through intentional forgetting tasks[42–46]. For example, using the direct forgetting paradigm[43,45] or Think/No-Think paradigm[46], it was found that children often failed to inhibit unwanted information that had become unnecessary for the current task. The critical difference between these studies and the present study is that, while these studies focus on the ability to remove outdated information that has already been stored in memory, the current study focuses on the ability to prevent outdated information from entering memory[22–25]. These abilities are thought as two important ways of how inhibitory mechanisms regulate the contents of memory[38]. Future studies can further explore whether the development of these two kinds of inhibition shares a common process, or diverges in certain aspects.

The retention of attended yet outdated information appears to impose a burden on the capacity-limited working memory, resulting in less efficient information processing. However, viewed from another perspective, it could serve as an adaptive function for children's learning. Firstly, the absence of selectivity implies a broader exploration of information, rather than a focus on specific subsets. This process has been recognized as a significant driver of cognitive development[47,48]. Additionally, retaining the past event in memory may facilitate the identification of sequential patterns, such as associative learning or causality inference, wherein one event follows another with a high probability. Notably, some studies revealed that children outperformed adults in grasping such relationships[49–51]. This could possibly be attributed to their over-retention of expired information.

## Methods
### Participants
We performed a priori power analysis using G*Power 3.1[52] to estimate the appropriate number of participants. The attribute amnesia effect size (φ) was estimated as 0.49 based on the results of Chen and Wyble[17]. The power calculation yielded an estimated minimum of 17 participants to detect such effect with 80% power (with α set to 0.05). We set the sample size as 20 for each age group in Experiments 1–4. The power analysis of Experiment 5 was based on the performance difference in the surprise test between adults and children in Experiments 1–4, which was φ = 0.39. To be conservative, we chose φ = 0.30 to estimate the sample size, yielding an estimated minimum of 44 participants in each group to detect such effect with 80% power (with α set to .05). We set the sample size as 50 for each age group in Experiment 5. The sampling procedure was convenience sampling. The adults were recruited from Zhejiang University in exchange for course credits or a monetary payment, and all of them reported normal or corrected-to-normal visual acuity. The children were recruited from a kindergarten in Hangzhou, China, with no reported vision, hearing, or developmental issues. The study was approved by the Institutional Review Board at the Department of Psychology and Behavioral Sciences, Zhejiang University. Informed consent was obtained from the parents of the children and the adult participants themselves prior to each experiment.

One adult and two children in Experiment 4 were excluded due to failed eye-tracking data recoding. One adult in Experiment 4 and one adult in Experiment 5 were excluded for poor performance ( < 60%) in pre-surprise trials. All removed participants were replaced by new valid participants to achieve the planed sample size in each experiment. Demographic information was collected through participant self-reporting of their age (in years) and sex (binary choice: male or female) prior to the commencement of the experiment. The mean age and sex characteristics of the participants in each experiment were as follows: Experiment 1, adults: $M_{age}$ = 18.95 ± 1.00 years (13 women and 7 men), children: $M_{age}$ = 5.75 ± 0.85 years (8 girls and 12 boys); Experiment 2, adults: $M_{age}$ = 20.50 ± 3.50 years (18 women and 2 men), children: $M_{age}$ = 6.10 ± 0.55 years (9 girls and 11 boys); Experiment 3, adults: $M_{age}$ = 24.20 ± 2.78 years (5 women and 15 men), children: $M_{age}$ = 5.95 ± 0.51 years (5 girls and 15 boys); Experiment 4, adults: $M_{age}$ = 19.80 ± 1.79 years (9 women and 11 men), children: $M_{age}$ = 6.00 ± 0.56 years (9 girls and 11 boys); Experiment 5, adults: $M_{age}$ = 21.76 ± 3.02 years (28 women and 22 men), children: $M_{age}$ = 5.48 ± 0.61 years (21 girls and 29 boys). There was no overlap in participants across the experiments.

## Stimuli and procedure

All experiments were programmed and executed using MATLAB software (The MathWorks; Natick, MA) with the Psychophysics Toolbox extension[53–55] and presented on a 14-inch laptop monitor (60 Hz, 1024 × 768 screen resolution). Participants sat at a viewing distance of approximately 50 cm. The background of the display was medium gray (RGB: 150, 150, 150). The adults were tested in a quiet room in Zhejiang University and the children were tested in a quiet room in the kindergarten. The investigators were not blinded to allocation during experiments and outcome assessment.

In Experiment 1, each trial began with a centered black fixation cross (0.6° of visual angle) among four black placeholder circles (0.6°). The four placeholders were presented on the four corners of an invisible square (5.2° × 5.2°) centered on the screen. After a variable duration (800-1800 ms), the stimulus array appeared for 250 ms. The stimulus array contained one English letter target (A, B, D, or E; 0.6° × 0.8°) and three Arabic number distractors (2 to 9; 0.6°×0.8°), which were randomly presented at the four locations of the placeholders. This stimuli display was followed by a 500-ms fixation cross display. In the first 11 pre-surprise trials, participants were asked to report the location where the target letter had appeared (location task). Feedback on the correct target location was given after the location task. In the 12th trial (i.e., the surprise trial), prior to the location task, participants were unexpectedly presented with a forced-choice question requiring them to indicate which of four letters was the target letter they had just seen. Following the surprise trial, participants completed four control trials which were in the same format as the surprise trial. The adults responded by pressing corresponding keys on a keyboard, whereas children made verbal responses that were recorded by the experimenter. There were eight practice trials with the same format as the pre-surprise trials but at a slower pace before the formal experiment.

In Experiment 2, we first selected eight common animal pictures (i.e., chicken, dog, frog, rabbit, cat, pig, turtle, goat), and then asked 20 additional children from the same kindergarten to select four animal pictures that they were most familiar with. The four selected most familiar animals (i.e., chicken, dog, rabbit, cat; the scoring table could be found at [https://osf.io/qhy3r/]) served as the targets in the attribute amnesia task. The attribute amnesia task was identical to Experiment 1, with the exception that participants were required to locate a target animal (chicken, dog, cat, or rabbit, ~2.8° × 2.8°) among three clothing distractors (coat, hat, pants, shoe, sweater, or boot, ~2.8° × 2.8°). The four locations of the stimulus array were distributed on the corners of an invisible square (7.8° × 7.8°). The stimuli were presented for 1000 ms and followed by a 400-ms blank screen. In the surprise test, participants were asked to report which animal they had just found. This experiment was pre-registered on Open Science Framework [https://osf.io/4hyj8] (March 18, 2023) and performed with no deviations from the pre-registration.

Experiment 3 was identical to Experiment 1 except as follows. Participants were asked to locate a target number that was larger than five (6, 7, 8, or 9) among three distractor numbers smaller than five (1, 2, 3, or 4). The stimulus presentation time was 1000 ms. In the surprise test, participants were asked to report the identity of the target number. For both groups of adults and children, the experimenter directly asked participants to indicate the target number in the surprise test.

Experiment 4 was identical to Experiment 3 except as follows. An EyeLink II system (EyeLink Portable Duo, SR Research, Mississauga, Ontario, Canada) recorded eye position monocularly from the left eye with a sample rate of 1000 Hz. At the beginning of the experiment, the eye tracker was calibrated using a 5-point calibration procedure. During the task, participants needed to fixate at the central black circle and press the space key to start a trial. The eye tracker was recalibrated if a participant failed to fixate the central circle at the beginning of the trial. For better recording of eye movement data, the four locations of stimulus array were distributed on the corners of a slightly larger

invisible square (7.8°×7.8°). The stimuli were presented for 1000 ms and followed by a 400-ms blank screen. There were 32 trials, including 27 pre-surprise trials, 1 surprise trial and 4 control trials. The adults responded by pressing corresponding keys on a keyboard, whereas children made verbal responses that were recorded by the experimenter.

Experiment 5 was identical to Experiment 2. This experiment was not pre-registered.

## Data analysis

For eye movement data analysis in Experiment 4, we focused on the fixation distribution during the stimulus array in pre-surprise trials. Trials with no eye movement or false responses in the location task were excluded. The stimulus area was defined as a circle around the stimulus center, which was ~3.3° larger than the actual stimulus size.

We used $\chi^2$ tests, two-tailed $t$ tests and ANOVAs for all statistical analyses. The normality test (Shapiro-Wilk) showed no significant deviation in normality for the eye movement data used for $t$ tests and ANOVAs. In ANOVAs, we reported the Greenhouse-Geisser-corrected $p$ value if sphericity was violated. We calculated the Bayes factors of the nonsignificant results to present how strongly the data supported the null hypothesis, using JASP Bayesian Contingency Tables (Prior concentration = 1, Sampling model = Independent multinomial)[56].

## Reporting summary

Further information on research design is available in the Nature Portfolio Reporting Summary linked to this article.

## Data availability

The datasets (including preprocessed data) generated in this study have been deposited on the Open Science Framework [https://osf.io/qhy3r/][57]. Source data are provided in this paper.

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

## Acknowledgements

This work was supported by grants from Science and Technology Innovation 2030-"Brain Science and Brain-like Research" Major Project (No.2022ZD0210800 for H.C.), Emerging Enhancement Technology under Perspective of Humanistic Philosophy, supported by National Office for Philosophy and Social Science (No. 20&ZD045 for H.C.), National Natural Science Foundation of China (No.32171046 for H.C., No.32371108 for J.H., No.32071044 for M.S., No.32200844 for Y.F.), China Postdoctoral Science Foundation (No.2022M712789 for Y.F.). We are very grateful to Dr. Wei (Sophia) Deng and Dr. Brad Wyble for their helpful comments on the manuscript. We also thank Bin Li, Yiling Zhou, all the teachers, and children in Xihu Zijing Kindergarten for their assistance in data collection.

## Author contributions

Y.F. was responsible for conceptualization, data curation, formal analysis, funding acquisition, investigation, methodology, visualization, writing-original draft, and writing-review and editing. T.G. was responsible for conceptualization, data curation, formal analysis, investigation, validation, writing-original draft, and writing-review and editing. J.Z. was responsible for investigation, validation, and writing-review and editing. J.H. and M.S. were responsible for conceptualization, funding acquisition, methodology, and writing-review and editing. H.C. was responsible for conceptualization, data curation, funding acquisition, methodology, supervision, and writing-review and editing.

## Competing interests

The authors declare no competing interests.
