## [Peer Review File · Nature Communications]

Children exhibit superior memory for attended but outdated information compared to adults.REVIEWER COMMENTS

Reviewer #1 (Remarks to the Author):

In this review, it is shown that young children (4-6 years I believe) do not show the same attribute amnesia as young adults do in a surprise test for items that were attended, but then no longer needed (as far as participants knew). This was the case even when the attribute was in fact needed, then no longer needed. This means that the children had better memory of the no-longer-needed information than adults did. This pattern persisted in Experiment 2 even when participants had to find a digit larger than 5 as the target, and therefore had to process the target identity. //

This is a very nice project. In most ways, it seems quite valid. I have only one concern, but I think that concern warrants another experiment. This phenomenon found in adults may depend on the categorization becoming somewhat automatic, with a consistent mapping of items to the target and non-target categories (see the Psychological Review 1977 paper by Shiffrin and Schneider, and the one that same year by Schneider and Shiffrin). For adults, even deciding if a digit is over 5 may become consistently mapped. For a young child, even the distinction between letters and numbers may still be more effortful. //

What I would like to see is an additional, nonverbal experiment in which, for sure, children and adults both could do the same consistent mapping. An example is, find the item that is an animal (with something like clothing as the non-target category). Then the surprise test is on which animal it was. If automaticity of the judgment is what is needed, then children should now perform like adults. If instead, inhibition of the no-longer-needed item is the key, then the present pattern of results should be repeated once more.

Reviewer #2 (Remarks to the Author):

This is a well written manuscript presenting three experiments examining "attribute amnesia" across development. The authors replicate previous findings of attribute amnesia in adults, while presenting novel finding pointing to a lack of attribute amnesia in 5-to-6-year-old children. The research is well conducted, and Experiment 2-3 replicate the main finding while eliminating alternative explanations. I am confident that the manuscript deserves to be published. I am less confident that it has enough novelty and theoretical impact to be published in Nature Communications. The authors try to link the results to developmental differences in "memory selectivity" that presumably operates on top of "attentional selectivity". While this explanation is possible, it is also possible that greater familiarity with stimuli allows adults to identify the target on the basis of its visual features, without accessing its identity. Experiment 2 somewhat undermines but not fully eliminates this possibility because all the numbers are drawn from the set of 1 to 9 which is highly familiar to adults. This differential novelty explanation is supported by two sets of previous findings. First, Chen & Howe (2017, PeerJ) demonstrated that "attribute amnesia" was substantially attenuated when stimuli were not repeated within an experiment. And second, Plebanek and Sloutsky (2017, Psychological Science, Experiment 2) demonstrated that when novel stimuli were used, memory for search targets was comparably accurate in children and adults (with no evidence for "feature amnesia"). At the same, children had a more accurate memory for search-irrelevant features (which suggests developmental differences in attentional selection). Therefore, it is possible that the reported results are based on differential novelty of stimuli in children and adults. If this is the case, the findings are more indicative of (a) differences in how novel and familiar stimuli are processed during visual search rather than of (b) developmental differences in memory selectivity. I think that both (a) and (b) are interesting possibilities, but they have vastly different theoretical consequences. And to fully establish these consequences one of the two has to be eliminated, which would require additional experiments.

REVIEWER COMMENTS

Reviewer #1 (Remarks to the Author):

In this review, it is shown that young children (4-6 years I believe) do not show the same attribute amnesia as young adults do in a surprise test for items that were attended, but then no longer needed (as far as participants knew). This was the case even when the attribute was in fact needed, then no longer needed. This means that the children had better memory of the no-longer-needed information than adults did. This pattern persisted in Experiment 2 even when participants had to find a digit larger than 5 as the target, and therefore had to process the target identity. //

This is a very nice project. In most ways, it seems quite valid. I have only one concern, but I think that concern warrants another experiment. This phenomenon found in adults may depend on the categorization becoming somewhat automatic, with a consistent mapping of items to the target and non-target categories (see the Psychological Review 1977 paper by Shiffrin and Schneider, and the one that same year by Schneider and Shiffrin). For adults, even deciding if a digit is over 5 may become consistently mapped. For a young child, even the distinction between letters and numbers may still be more effortful. //

What I would like to see is an additional, nonverbal experiment in which, for sure, children and adults both could do the same consistent mapping. An example is, find the item that is an animal (with something like clothing as the non-target category). Then the surprise test is on which animal it was. If automaticity of the judgment is what is needed, then children should now perform like adults. If instead, inhibition of the no-longer-needed item is the key, then the present pattern of results should be repeated once more.

Response: Thank you very much for your positive feedback on our work and for suggesting this intriguing experiment. Based on your suggestion, we conducted a new experiment (Experiment 2 in the revised manuscript) using animal pictures that were highly familiar to both adults and children. In this experiment, participants were asked to locate a target animal among clothing, and then in a surprise test, they were asked to report which animal they had just found. To ensure that the children were indeed familiar with the animals, we had asked 20 additional children from the same kindergarten to select four animal pictures that they were most familiar with, out of eight common animals (chicken, dog, frog, rabbit, cat, pig, turtle, and goat). The four most familiar animals (chicken, dog, rabbit, and cat) were then used as targets in the attribute amnesia task for both the children and adults. The results of this new experiment replicated previous findings, with 11 out of 20 adults (55% correct)

and 17 out of 20 children (85% correct) accurately reporting the target identity in the surprise trial. Comparing the performance of adults and children, the results showed that children performed significantly better than adults in reporting the target animal (55% vs. 85%), with $\chi^2(1, N=40)=4.286, p=0.038, \phi=0.327$. These results suggest that even when using animal pictures that are highly familiar to both adults and children, difference in attribute amnesia still exists between the two groups.

Furthermore, as indicated by the research of Shiffrin and Schneider (1977) and Schneider and Shiffrin (1977), it is possible for participants to learn the consistent mapping of items to target and non-target categories. Nonetheless, their findings also suggested that thousands of training trials would be required for participants to establish categories of target and distractor sets when the stimuli are of the same type (for example, using Letter Set 1 as targets and Letter Set 2 as distractors). Given that our original Experiments 2 and 3 used stimuli from the same category (i.e., numbers) as both targets and distractors, and with only 10-30 trials in our brief experiment, it seems unlikely that participants could have acquired such a mapping. In fact, we conducted two additional control experiments (Experiments 4a and 4b) which demonstrated that participants did indeed access the identity/magnitude of the target number in the specific task set utilized in our study, rather than relying on categorization. For a more comprehensive description, please refer to our response to Reviewer 2.

Reviewer #2 (Remarks to the Author):

This is a well written manuscript presenting three experiments examining “attribute amnesia” across development. The authors replicate previous findings of attribute amnesia in adults, while presenting novel finding pointing to a lack of attribute amnesia in 5-to-6-year-old children. The research is well conducted, and Experiment 2-3 replicate the main finding while eliminating alternative explanations. I am confident that the manuscript deserves to be published. I am less confident that it has enough novelty and theoretical impact to be published in Nature Communications. The authors try to link the results to developmental differences in “memory selectivity” that presumably operates on top of “attentional selectivity”. While this explanation is possible, it is also possible that greater familiarity with stimuli allows adults to identify the target on the basis of its visual features, without accessing its identity. Experiment 2 somewhat undermines but not fully eliminates this possibility because all the numbers are drawn from the set of 1 to 9 which is highly familiar to adults. This differential novelty explanation is supported by two sets of previous findings. First, Chen & Howe (2017, PeerJ) demonstrated that “attribute amnesia” was substantially attenuated when stimuli were not repeated within an experiment. And second, Plebanek and Sloutsky (2017, Psychological Science, Experiment 2) demonstrated that when novel stimuli were used, memory for search targets was comparably accurate in children and adults (with no evidence for “feature amnesia”). At the same, children had a more accurate memory for search-irrelevant features (which suggests developmental differences in attentional selection). Therefore, it is possible that the reported results are based on differential novelty of stimuli in children and adults. If this is the case, the findings are more indicative of (a) differences in how novel and familiar stimuli are processed during visual search rather than of (b) developmental differences in memory selectivity. I think that both (a) and (b) are interesting possibilities, but they have vastly different theoretical consequences. And to fully establish these consequences one of the two has to be eliminated, which would require additional experiments.

Response: Thanks for the fantastic feedback and suggestions. We fully concur that it's crucial to ensure that our findings accurately reflect developmental discrepancies in memory selectivity, rather than differences in how novel and familiar stimuli are processed during visual search. In fact, Reviewer 1 brought up a similar concern.

To address this concern, conducting an experiment that employs stimuli with comparable levels of familiarity (either novel or high familiar) for both adults and children would be highly compelling. As you've mentioned, using novel stimuli could sometimes weaken the amnesia effect even for adults (Chen & Howe,

2017; Plebanek & Sloutsky, 2017; but see Tam et al., 2021). This may be due to the fact that novel stimuli tend to capture more attention and elicit greater activation, making them more difficult to be actively suppressed or inhibited. It's worth noting that previous research has suggested that active inhibition is most effective for moderately activated information, rather than information with very high or low activation levels (e.g., Fu et al., 2021; Ritvo et al., 2019).

Hence, as suggested by Reviewer 1, we conducted a new experiment (Experiment 2 in the revised manuscript) using stimuli (animal pictures) that were highly familiar to both adults and children. In this experiment, participants were asked to locate a target animal among clothing, and then in a surprise test, they were asked to report which animal they had just found. To ensure that the children were indeed familiar with the animals, we had asked 20 additional children from the same kindergarten to select four animal pictures that they were most familiar with, out of eight common animals (chicken, dog, frog, rabbit, cat, pig, turtle, and goat). The four most familiar animals (chicken, dog, rabbit, and cat) were then used as targets in the attribute amnesia task for both the children and adults. The results of this new experiment replicated previous findings, with 11 out of 20 adults (55% correct) and 17 out of 20 children (85% correct) accurately reporting the target identity in the surprise trial. Comparing the performance of adults and children, the results showed that children performed significantly better than adults in reporting the target identity (55% vs. 85%), with $\chi^2(1, N=40)=4.286, p=0.038, \phi=0.327$. These results suggest that even when using animal pictures as stimuli that are highly familiar to both adults and children, difference in attribute amnesia still exists between the two groups.

Moreover, in order to further ensure that our findings reflect developmental differences in memory selectivity, it is crucial to ascertain that both adults and children accessed the target identity, rather than relying on categorization. While it is clear that children accessed the target identity in the task, as evidenced by their accurate reporting of the target number in the surprise trial, we need to demonstrate that adults also accessed the identity/magnitude of the target number. In the original manuscript (Experiments 2 and 3), we employed a task where participants were required to locate a target number larger than five among three distractor numbers smaller than five, with the aim of eliminating the possibility of categorization. We believe that this task necessitates accessing the identity of the target since previous studies have validated that the number comparison task demands direct access to numerical quantity for both adults and young children (e.g., Dehaene, 1996; Temple & Posner, 1998). Nonetheless, it would be more compelling to demonstrate that adult participants also accessed the identity/magnitude of target number in the current study's particular task set. Therefore, we conducted two additional control experiments (Experiments 4a & 4b in the revised manuscript) to provide

direct evidence that adults did access the target identity (rather than rely on categorization) in the current task. In Experiment 4a, participants were asked to judge whether a presented number (Arabic numbers between 1 and 9, except for 5) was larger or smaller than 5. We used the classical *distance effect* to evaluate the quantitative representation of numbers, which refers to the observed phenomenon that it is more difficult to compare two numbers if they are close than if they are far apart (Moyer & Landauer, 1967). Accordingly, if the quantitative information is represented in this task, then RT will decrease as the numerical distance (1–4 units) from the reference number (i.e., 5) increases. For instance, the reaction to the number 9 would be faster than the reaction to the number 6 among participants, because 9 is further away from the reference number 5, making it easier for the participants to judge. Otherwise, if participants complete this task through categorization without identifying the magnitude of number, no such distance effect should be observed.

As shown in Figure 1, we observed a clear distance effect. A one-way repeated-measures analysis of variance yielded a significant main effect of the numerical distance (1-4 units) from the reference number (i.e., 5), $F(2.144,40.745)=34.822$, $p<0.001$, $\eta_p^2=0.647$. Importantly, there was a significant linear trend where RT decreased as numerical distance increased, $F(1,19)=82.879$, $p<0.001$, $\eta_p^2=0.814$. This linear trend was consistent for both numbers smaller than five ($F(1,19)=19.677$, $p<0.001$, $\eta_p^2=0.509$) and numbers larger than five ($F(1,19)=22.617$, $p<0.001$, $\eta_p^2=0.543$). To further confirm that numerical quantity was accessed in the current study's particular task set, we measured the distance effect in Experiment 4b, where participants were asked to locate a target number larger than 5 among three distractor numbers smaller than 5 (a similar task as in original Experiments 2 and 3). As shown below, we again found a clear distance effect in this task set. There was a significant main effect of the numerical distance (1–4 units) from the reference number (i.e., 5), $F(2.115,40.193)= 8.950$, $p<0.001$, $\eta_p^2=0.320$, and a significant linear trend where RT decreased as numerical distance increased, $F(1,19)=32.530$, $p<0.001$, $\eta_p^2=0.631$. These results yielded strong evidence that the quantitative representation of the target number was activated in this task set.

In summary, these new experiments provided converging evidence that participants accessed the target identity even for the stimuli they were very familiar with (e.g., numbers for adults and animals for children) in the current task set, suggesting that the results reflected developmental disparities in memory selectivity, rather than variations in the familiarity of stimuli across different groups.

Figure 1: RT results of Experiments 4a and 4b. Error bars represent standard error of the mean.

References

- Chen, W., & Howe, P. D. (2017). Attribute amnesia is greatly reduced with novel stimuli. *PeerJ*, 5, e4016.
- Dehaene, S. (1996). The organization of brain activations in number comparison: Event-related potentials and the additive-factors method. *Journal of Cognitive Neuroscience*, 8(1), 47-68.
- Fu, Y., Zhou, Y., Zhou, J., Shen, M., & Chen, H. (2021). More attention with less working memory: The active inhibition of attended but outdated information. *Science Advances*, 7(47), eabj4985.
- Moyer, R. S., & Landauer, T. K. (1967). Time required for judgements of numerical inequality. *Nature*, 215(5109), 1519-1520.
- Plebanek, D. J., & Sloutsky, V. M. (2017). Costs of selective attention: When children notice what adults miss. *Psychological Science*, 28(6), 723-732.
- Ritvo, V. J., Turk-Browne, N. B., & Norman, K. A. (2019). Nonmonotonic plasticity: how memory retrieval drives learning. *Trends in Cognitive Sciences*, 23(9), 726-742.
- Schneider, W., & Shiffrin, R. M. (1977). Controlled and automatic human information processing: I. Detection, search, and attention. *Psychological Review*, 84(1), 1-66.
- Shiffrin, R. M., & Schneider, W. (1977). Controlled and automatic human information processing: II. Perceptual learning, automatic attending and a general theory. *Psychological Review*, 84(2), 127-190.
- Tam, J., Mugno, M. K., O'Donnell, R. E., & Wyble, B. (2021). And like that, they were gone: A failure to remember recently attended unique faces. *Psychonomic Bulletin & Review*, 28(6), 2027-2034.
- Temple, E., & Posner, M. I. (1998). Brain mechanisms of quantity are similar in 5-year-old children and adults. *Proceedings of the National Academy of Sciences*, 95(13), 7836-7841.

REVIEWER COMMENTS

Reviewer #1 (Remarks to the Author):

The authors have done an excellent job of considering the reviewers' concerns and I am enthusiastic about the resulting publication. I have no further comments.

Reviewer #2 (Remarks to the Author):

The revised manuscript addresses my previous concerns by conducting additional experiments. In my view, added Experiments 2 and 4 substantially strengthen the manuscript, making it worthy of publication in Nature Communications. My only suggestion is to move Experiment 4 to Supplementary Materials, while mentioning it in the main text. This is because the experiment, while wonderfully executed and necessary, serves a function of eliminating alternative explanations. I also suggest that the authors may want to mention that (at least in Experiments 1, 2, and 3), children, while succeeding on reporting identity on the surprised trial, utterly failed with location. Given that they were accurate in reporting location in the pre-surprise trials, this finding suggests that the act of reporting the identity impaired their memory for location. This seems like an interesting finding, but the authors could consider this a distraction and avoid mentioning it. Overall, I would like to congratulate the authors on the outstanding work!

REVIEWER COMMENTS

Reviewer #1 (Remarks to the Author):

The authors have done an excellent job of considering the reviewers' concerns and I am enthusiastic about the resulting publication. I have no further comments.

Thanks for your enthusiasm and all the previous comments that have significantly enhanced the quality of the paper.

As the editor mentioned in the decision letter, there remains an additional comment from you regarding the strength of the evidence presented in Experiment 2. Following the specific suggestions from the editorial team, we conducted a replication with a larger sample size of 50 participants in each group. This increased sample size was determined through a more conservative estimation of the effect size ($\varphi=0.30$) regarding the performance difference between children and adults in reporting the key feature during the surprise test. Fifty new adults ($M_{\text{age}} = 21.76$ years) and 50 new children ($M_{\text{age}} = 5.48$ years) participated in this replication experiment.

The new experiment exactly replicated the previous results. As shown in Table 1, only 23 of 50 (46% correct) adults were correct on the identity report task in the surprise trial, which significantly improved in the first control trial (46% vs. 98%), $\chi^2(1, N=100)=33.532, p<0.001, \varphi=0.579$. In contrast, 43 of 50 (86% correct) children correctly reported the identity of the target in the surprise trial, which showed no statistically significant difference from that in the first control trial (86% vs. 94%), $\chi^2(1, N=100)=1.778, p=0.182, \varphi=0.133, BF_{10}=0.343$; as well as from that in the following control trials (86% vs. 94%; 88%; 94%), $\chi^2(1, N=100)\leq 1.778, p_s\geq 0.182, \varphi\leq 0.133, BF_{10}\leq 0.343$. Importantly, as previously found, the between-group comparison showed that children performed significantly better than adults in reporting the target identity in the surprise trial (86% vs. 46%), $\chi^2(1, N=100)=17.825, p<0.001, \varphi=0.422$. These results confirmed that our previous findings were replicable and statistically reliable.

Table 1. Replication of Experiment 2 ($N=50$ for each age group)

	Pre-surprise	Surprise	Control1	Control2	Control3	Control4
Children						
Location	99%	58%	76%	74%	84%	86%
Identity	N/A	86%	94%	94%	88%	94%
Adults						
Location	99%	94%	98%	100%	98%	100%
Identity	N/A	46%	98%	100%	100%	100%

Note. N/A= Not applicable.

We included this replication as Experiment 5 in the revised manuscript, on Pages 16-18.

Reviewer #2 (Remarks to the Author):

The revised manuscript addresses my previous concerns by conducting additional experiments. In my view, added Experiments 2 and 4 substantially strengthen the manuscript, making it worthy of publication in Nature Communications. My only suggestion is to move Experiment 4 to Supplementary Materials, while mentioning it in the main text. This is because the experiment, while wonderfully executed and necessary, serves a function of eliminating alternative explanations. I also suggest that the authors may want to mention that (at least in Experiments 1, 2, and 3), children, while succeeding on reporting identity on the surprised trial, utterly failed with location. Given that they were accurate in reporting location in the pre-surprise trials, this finding suggests that the act of reporting the identity impaired their memory for location. This seems like an interesting finding, but the authors could consider this a distraction and avoid mentioning it. Overall, I would like to congratulate the authors on the outstanding work!

Thanks for your enthusiasm and all the helpful suggestions. As suggested, we have moved Experiment 4 to Supplementary Materials and mentioned it in the main text.

We also appreciate the reviewer for encouraging us to mention the potential interesting findings regarding the children's worse performance on location in the surprise trial, which was also found in our new replication experiment with a larger sample size. This interesting finding revealed that the surprise test of identity information elicited more disruption of location memory in children, indicating that children's working memory storage might be more likely to be interfered by unexpected events. We have included a brief discussion regarding this interesting finding on Page 18 in the revised manuscript.

REVIEWERS' COMMENTS

Reviewer #1 (Remarks to the Author):

I had previously recommended acceptance but the editors asked me to take a look at additional requests that were made to the authors. I think the response to these requests is fine. In particular, I was especially asked to attend to two issues. My answers are as follows. (1) Looking at an age x experimental phase interaction in terms of the age effect in the amount of improvement in the two groups is a perfectly acceptable way to examine the interaction. (2) Some of the Bayes Factors are indeterminate but I agree with the authors that their conclusions do not hinge on whether these particular effects are weakly reliable, indeterminate, or demonstrably null. I still recommend acceptance.

Reviewer #2 (Remarks to the Author):

The paper was very good from the very beginning, but it is striking how much it has improved in the course of the review process. Remarkable work, and I cannot wait to see it in press.

REVIEWER COMMENTS

Reviewer #1 (Remarks to the Author):

I had previously recommended acceptance but the editors asked me to take a look at additional requests that were made to the authors. I think the response to these requests is fine. In particular, I was especially asked to attend to two issues. My answers are as follows. (1) Looking at an age x experimental phase interaction in terms of the age effect in the amount of improvement in the two groups is a perfectly acceptable way to examine the interaction. (2) Some of the Bayes Factors are indeterminate but I agree with the authors that their conclusions do not hinge on whether these particular effects are weakly reliable, indeterminate, or demonstrably null. I still recommend acceptance.

Thanks for your enthusiasm. These valuable comments have greatly contributed to improving the quality of the paper.

Reviewer #2 (Remarks to the Author):

The paper was very good from the very beginning, but it is striking how much it has improved in the course of the review process. Remarkable work, and I cannot wait to see it in press.

Thanks for your enthusiasm and all the previous comments that have significantly enhanced the quality of the paper.